# Factors Associated with Mental Health Outcomes in Hospital Workers during the COVID-19 Pandemic: A Mixed-Methods Study

**DOI:** 10.3390/ijerph19095346

**Published:** 2022-04-28

**Authors:** Jeff Huarcaya-Victoria, Beltran Villarreal-Rao, Matilde Luna, Wendoline Rojas-Mendoza, Christoper A. Alarcon-Ruiz, David Villarreal-Zegarra, Ana L. Vilela-Estrada, Samantha Ramírez

**Affiliations:** 1Departamento de Psiquiatría, Hospital Nacional Guillermo Almenara Irigoyen, EsSalud, Lima 15033, Peru; beltran.villarreal@essalud.gob.pe (B.V.-R.); matildelunamatos@gmail.com (M.L.); 2Departamento de Salud Mental, Hospital Nacional Edgardo Rebagliati Martins, EsSalud, Lima 15072, Peru; wendy037@gmail.com; 3Neurociencia, Efectividad Clínica y Salud Pública, Universidad Científica del Sur, Lima 15067, Peru; calarconru@cientifica.edu.pe; 4Dirección de Investigación, Desarrollo e Innovación, Instituto Peruano de Orientación Psicológica, Lima 15046, Peru; davidvillarreal@ipops.pe (D.V.-Z.); vilelaestradaana@ipops.pe (A.L.V.-E.); 5Independent Researcher, Lima 15001, Peru; sam.ramirez.castillo@gmail.com

**Keywords:** anxiety, stress, psychological, depression, COVID-19, health personnel, Peru

## Abstract

Background: We investigated some of the factors associated with depression, perceived stress, and anxiety in clinical and nonclinical healthcare workers of two hospitals. Methods: A mixed-methods approach was used. The sample included clinical (physicians, nurses, and others) and nonclinical (security and cleaning staff) healthcare workers of two tertiary hospitals in Peru. Participants completed an online self-survey. In the qualitative analysis, data were subjected to thematic analysis. Results: We analyzed data from 613 participants, of which 8.6%, 9.0%, and 78.2% had moderate-to-severe anxiety, depression, and perceived stress, respectively. Having a previous mental health problem, being concerned about losing one’s job, having at least two COVID-19 symptoms in the preceding two weeks, and being afraid of infecting family members increased the prevalence of experiencing moderate-to-severe depression and anxiety. The qualitative analysis allowed us to identify five recurring factors that caused a negative impact on workers’ lives during the pandemic: emotional distress linked to hospital experiences of suffering and death, modification of routines, fear of COVID-19, exacerbation of mental disorders, and physical problems associated with emotional distress. Conclusions: Clinical and nonclinical healthcare workers in Peru have experienced depression, anxiety, and stress during the COVID-19 pandemic. Future research and interventions are necessary to improve psychological support for hospital workers.

## 1. Introduction

The World Health Organization has categorized the pandemic of the novel coronavirus SARS-CoV-2 disease, COVID-19, as a global health emergency [1]. On 1 April 2022, the worldwide count stood at 486,761,597 confirmed cases and 6,142,735 deaths [2]. At the time, Peru had 3,544,862 confirmed cases and 212,102 deaths, which is among the highest rate of cases in the world, and had a case lethality rate of 5.98% [3]. During the pandemic, negative emotions and thoughts have become more common due to the high infection rate and the increasing number of cases and deaths from COVID-19 [4]. This is a threat to the population’s mental health, because it causes mental health issues such as anxiety, depression, and stress [5]. The COVID-19 pandemic is a new type of trauma that has never been conceptually or empirically analyzed in the field of mental health and psychiatry research [6]. Several factors make the COVID-19 pandemic a unique type of trauma: (a) it is a constant and permanent traumatic stress; (b) it is a multiple complex trauma (COVID-19 traumatic stress consists of various components: fear of the present and future infection and death, actual economic hardship, stressors related to lockdown, etc.); and (c) it is not necessarily related to the actual infection of COVID-19, but also is more related to the perceived threat of the uncontrolled virus and the direct and indirect social consequences [6]. The COVID-19 pandemic can be understood as a traumatic stressor capable of exacerbating mental health problems [7].

### 1.1. Mental Health in Healthcare Workers of COVID-19 Patients

Although the healthcare systems of most countries have been affected by the pandemic, some population groups may be even more vulnerable to its detrimental effects on mental health than others [8,9]. Peru is no exception to this increased mental health burden and greater psychosocial reactions during the COVID-19 pandemic [10]. This burden disproportionately affects women, people with low income, and young people [10]. 

Essential workers conduct a range of functions that are necessary to ensure the continued viability of critical infrastructure [11]. The essential workers who are directly or indirectly involved in the healthcare of COVID-19 patients, especially clinical healthcare workers, have a higher risk of developing mental symptoms [12,13]. 

### 1.2. Factors Associated with Mental Health in Healthcare Workers

Several factors may contribute to the appearance of mental symptoms in healthcare workers: having a significantly higher workload due to the risk of infection, a lack of adequate personal protection, isolation, increasing work demands, frustration, fatigue from long working hours, little to no contact with their families, and continued proximity with patients expressing negative emotions [14]. Other factors, such as fear of infection, fear of quarantine, and feelings of betrayal by their employers may cause a significantly negative impact on healthcare workers’ mental health [15]. 

Stigmatization is common in disease outbreaks and pandemic situations. Social stigma against healthcare workers who are taking care of COVID-19 patients is expected [16]. Frontline workers experienced three times more stigma than those who did not work on the frontline [17]. The impact of stigmatization against healthcare workers is not limited to their psychological wellbeing; it can also affect their professional competencies to provide high-quality care to the patients during the pandemic [16].

Previous studies have found that perceived job insecurity has consequences not only on the individuals’ financial capacity, but also in their mental health [18,19]. Employment uncertainty causes fear of poverty and leads to marginalization, stigmatization, and social exclusion [20]. The presence of job insecurity is a strong predictor for depression [21]. 

Additionally, nonclinical healthcare workers may have lower capacity for having contact with COVID-19 patients [22] while being exposed to the same factors as clinical workers [23]. A lack of balance among professional duty, altruism, and constant fear can cause conflicts and cognitive dissonance in many hospital workers [24]. The identified risk and protective factors were summarized as intrapersonal (sex, age, sleep quality, etc.); interpersonal (marital status, familial support, family history of mental disease, peer support, etc.); and organizational (years of work experience, level of training, job satisfaction, etc.) [25]. In addition to emotional coping strategies, religious faith had a positive impact on the mental health and level of happiness of healthcare workers during the COVID-19 pandemic [26]. A meta-analysis reported a prevalence of anxiety symptoms (26%), depressive (25%), and post-traumatic stress disorders (3%) in medical staff [27]. These changes impacted the mental wellbeing of workers and their work activity, their attention in patients, and their decision-making process [14]. Consequently, if suppression of the pandemic is the target goal, hospital workers’ mental wellbeing must be acknowledged [14]. 

### 1.3. Aims of the Investigation

Our immediate priority was to collect data of the psychological effects, caused by the COVID-19 pandemic, on vulnerable groups such as healthcare workers [28]. However, nonclinical hospital workers are usually not included in scientific publications about the COVID-19 pandemic outcomes. Furthermore, based on the need to respond to the crisis, many studies focused on epidemiological and quantitative studies [13,29]; however, less attention was put on the contextualized experiences and meanings attributed to COVID-19, and to the strategies to mitigate its spread on healthcare and non-healthcare workers of hospitals [30]. Quantitative data analysis can provide a more accurate understanding about the experience of people; thus, by combining the quantitative and qualitative methods of the self-reported effects of the COVID-19 pandemic on the mental health of hospital workers we will be able to obtain a clearer picture of the issue. 

Due to the lack of information about the impact of COVID-19 on Peruvian hospital workers’ mental health, we developed a sequential quantitative-qualitative mixed-methods study design to examine the following research questions: 1. What is the prevalence of mental health problems in a sample of workers from two tertiary hospitals in Lima, Peru? 2. What factors are associated with positive or negative mental health outcomes in this sample? 3. What are the workers’ perceptions and beliefs about the impact of the pandemic on their mental health? 

## 2. Materials and Methods

### 2.1. Study Design

We performed primary analysis of a dataset, which included clinical and nonclinical workers of two tertiary hospitals in Lima, Peru. Participants were surveyed by using an online self-survey. We collected the data during the initial phase of the COVID-19 pandemic, from July to September 2020.

### 2.2. Setting

Edgardo Rebagliati Martins National Hospital and Guillermo Almenara Irigoyen National Hospital are tertiary hospitals located in Lima, Peru, and they are the two most important health centers in Peru. Both hospitals belong to the Social Security health system network (EsSalud), one of the four healthcare systems of Peru. EsSalud is financed by the Ministry of Labor, providing healthcare to formal current and former workers, and their relatives [31]. During the COVID-19 pandemic, both hospitals were infection hotspots, having several COVID-19 patients with moderate-to-severe levels of the disease.

During the peak caseload in Peru in August 2020, these hospitals had a massive overload of patients, which caused scarcity of personal protective equipment, medicinal oxygen, beds in intensive care units, and mechanical ventilators. This forced clinical and nonclinical workers to make 12-, 24-, or even 36-h shifts in the hospitals. Clinical workers (in particular, physicians, nurses, nurses’ technicians, and others) were responsible for treating the patients, most of whom had COVID-19. Nonclinical healthcare workers are generally security personnel and cleaning staff inside the hospitals. Security personnel keep order both outside and inside the facility, and the cleaning staff work in all hospital areas and oversees common and medical waste management. Both security and cleaning staff are nonformal workers, and a third-party company that provides services to the hospital hires them. All workers inside the hospital are considered essential healthcare workers during the pandemic [32].

### 2.3. Participants

From 27 July to 6 September 2020, we surveyed a nonprobabilistic sample of clinical (specialists physicians, resident physicians, general physicians, nurses, technicians, nutritionists, psychologists, and midwives) and nonclinical (security and cleaning staff) healthcare workers at Edgardo Rebagliati Martins National Hospital and Guillermo Almenara Irigoyen National Hospital. We included all the workers who were working at the hospitals during at the time of the study and had been working at the hospital for the previous 3 months. The areas where the clinical and nonclinical healthcare workers labor included the emergency department, inpatient departments, intensive care units, and administrative areas. We excluded those participants who did not indicate any of the three primary outcomes (depressive symptoms, anxiety symptoms, or perceived stress) and those who did not complete the survey.

### 2.4. Procedures

The complete online survey and informed consent were provided through the SurveyMonkey platform. We sent a link of the survey to workers in both hospitals using a snowball method. In addition, we contacted at least one healthcare worker from each area (the emergency department, inpatient areas, intensive care units, and administrative areas) and at least one worker from both security and the cleaning staff from each hospital to ask them to distribute the webpage link to the workmates of their respective areas. The survey was self-administered. The participants were allowed to respond to the survey at any time. The webpage link was distributed to the participants through messages on WhatsApp and Facebook private groups.

### 2.5. Instruments

#### 2.5.1. Perceived Stress

We used the Spanish version of the Perceived Stress Scale-10 (PSS-10), validated in Colombia, to measure perceived stress over the previous two weeks [33]. The PSS-10 is the best-known instrument for quantifying emotional stress in clinical and epidemiological research. Each of the scale’s 10 items (i.e., “Have you been upset because of something that happened unexpectedly?”) is assessed on a 5-point Likert scale from 0 (Never) to 4 (Very often). The scores of all the items are totaled and categorized into three levels of perceived stress: mild (0–13), moderate (14–26 score), and severe (27–40 score) [34]. The PSS-10 had a Cronbach’s alpha value of 0.65, a McDonald’s omega value of 0.68, and a Mosier coefficient of 0.68 [33].

#### 2.5.2. Depressive Symptoms

We used the Patient Health Questionnaire-9 (PHQ-9) to evaluate depressive symptoms. This is a self-administered scale consisting of nine items that assess depression (i.e., “Little interest or pleasure in doing things?”). The items are assessed on a 4-point Likert scale ranging from 0 (never) to 3 (almost every day). The item scores are summed, and the depressive symptoms are categorized into five categories: normal (0–4), mild (5–9), moderate (10–14 score), moderate-to-severe (15–19 score), and severe (20–27 score). A score of ≥10 has been recommended as the cut-off score for detecting clinically significant major depressive disorders [35]. Studies that use the PHQ-9 in Latin America have identified this as a valid and reliable tool for detecting depressive symptoms in various populations [36,37]. A Spanish version of the PHQ-9 has been validated in Peru. The validated features included the internal structure, measurement invariance, and internal consistency [38].

#### 2.5.3. Anxiety Symptoms

We used the Generalized Anxiety Disorder-7 (GAD-7) instrument to evaluate anxiety symptoms. This is a valid and efficient self-administered scale adopted to assess the severity of anxiety disorders in clinical practice [39]. It consists of seven items (i.e., “Feeling nervous, anxious, or on edge”) with responses on a 4-point Likert scale, ranging from 0 (never) to 3 (almost every day). Responses are summed and used to categorize anxiety symptoms into four categories: normal (0–4 score), mild (5–9 score), moderate (10–14 score), and severe (15–21 score). A score of ≥10 was been recommended as the cut-off score for indicating clinically significant anxiety [39]. The content validity and the relevance and adequacy of the items in the Spanish cultural context were confirmed in an earlier study [40]. An assigned cut-off value of 10 showed adequate values of sensitivity (86.8%) and specificity (93.4%) [40].

#### 2.5.4. Other Variables

We collected sociodemographic variables such as age, sex (female or male), marital status (single, married or living together, divorced, or widowed), religious (yes or no), history of a diagnosed mental health problem (yes or no), living alone (yes or no), living with a person at risk for COVID-19 (yes or no), and occupational information, including working area/department and position. 

Other variables collected were the perceived stigma of working at a hospital, fear of infecting family members, and fear of losing employment. To answer these questions, the participant used a 4-point Likert scale as follows: 1, no days; 2, several days (between 1–6 days); 3, more than half of the days (between 7 and 11 days); and 4, almost every day (12 or more days). Subsequently, these variables were dichotomized: the first two options were grouped as “Less than half of the days over the previous two weeks,” and the last two were grouped as “More than half of the days over the previous two weeks.” Finally, we asked participants to choose from a list of several symptoms related to COVID-19, which did they feel once or more over the previous two weeks. This last variable was dichotomized into having two or more symptoms of COVID-19 over the previous few weeks or not. 

### 2.6. Analysis

We compiled the data into an anonymized database without duplicates. For the descriptive analysis, we reported absolute and relative frequencies for all categorical variables. Before further analysis, we dichotomized all mental health outcomes: Perceived stress was divided into mild (0–13) and moderate-to-severe (14–41); depression symptoms into normal-to-mild (0–9) and moderate-to-severe (10–27); and anxiety symptoms into normal-to-mild (0–9) and moderate-to-severe (10–21). We considered that moderate-to-severe symptoms are clinically relevant for the patients and they may require mental health support. Subsequently, for the bivariable analysis of each mental health outcome and its covariates, we used the chi-2 test.

We used generalized linear models with the Poisson family, link log function, and clustered by the hospital to calculate the raw (rPR) and adjusted prevalence ratios (aPR), and the 95% confidence intervals (95% CIs) between each covariate and the three dichotomized mental health outcomes. A manual stepwise-forward regression was used to test which covariates were independently associated to the three mental health outcomes, and it was determined to be included into the multivariate model. We used the likelihood-ratio test to compare each covariate model with the null model. The covariate with a minor *p*-value in the likelihood-ratio test was added to the multivariable model. This process was repeated until there were no covariates with *p* < 0.05 in the likelihood-ratio test. Each mental health outcome was independently assessed. Data analysis was performed using Stata software version 16.0 (STATA Corporation, College Station, TX, USA). We considered *p* < 0.05 to indicate statistical significance.

### 2.7. Qualitative Analysis

We conducted a qualitative interview survey with a phenomenological-hermeneutical approach to identify the impact of COVID-19 on health personnel.

In sum, 20 participants were selected from the quantitative study through an intentional sampling, considering as inclusion criteria a high score for anxiety symptoms, depressive symptoms, or stress. Oral informed consent for the study was obtained before each interview, as well as for recording and anonymizing the transcription. The interviews were conducted via video or audio call in October 2020.

We carried out a textual transcription of the interviews, and we descriptively encoded the categories and subcategories by using the Atlas.ti 7.5.18 software (Scientific Software Development GmbH, Berlin, Germany), which allowed discourse ordering. A group of research specialists in psychology, psychiatry, medicine, and nursing who were familiar with qualitative research, examined the collected the information using a thematic analysis strategy. We identified five recurring topics related to changes in workers’ lives at a personal level: emotional distress linked to hospital experiences of suffering and death, exacerbation of mental disorders, physical problems associated with emotional distress, modification to life routines, and fear of COVID-19. We collected and analyzed the data in Spanish, and translated the obtained results into English.

### 2.8. Ethics

All the participants volunteered their involvement and provided informed consent before the study began. This informed consent indicated that neither participation, nor refusal, nor abbreviated participation would have consequences on their work or position. Furthermore, we offered psychiatric help for any participant willing to accept it. We offered no economic compensation. We previously registered the research protocol in the PRISA platform, an online registry administered by Peruvian National Health Institute, specialized on COVID-19 and tuberculosis research (ID code: EI00000001342). The COVID-19-specific Institutional Review Board approved the research protocol for the data collection and data analysis on EsSalud’s institutions (Seguro Social del Perú, Lima).

## 3. Results

### 3.1. Quantitative Results

#### 3.1.1. Participants

The original database included 1139 responses to the online questionnaire. After applying the inclusion criteria, data from 613 participants remained for the analysis (Figure 1). Most of the participants were women (61.8%), single (50.2%), living with family members (86.1%), and nonclinical workers (50.2%). Other sociodemographic characteristics can be observed in Table 1.

We identified a high incidence of mental health problems in the participants. 28.2% of participants were identified as having mild, moderate, or severe anxiety symptoms; 29.2% had mild, moderate or severe depressive symptoms; and 78.5% had moderate or severe perceived stress levels (Table 1).

#### 3.1.2. Bivariate Association

Living with a person at risk of COVID-19, having a history of mental health problems, feeling stigmatized for working at a hospital, being afraid of infecting family members, thinking about losing employment, and having two or more symptoms of COVID-19 in the previous two weeks are all associated with increased anxiety, depression, and perceived stress scores. Professing a religion, being a clinical healthcare worker, being older, and being a woman were associated with greater depressive symptoms but not a greater level of anxiety or stress (Table 2).

#### 3.1.3. Regression Model

The raw models indicated that feeling stigmatized for working at a hospital was among the factors that most increased the prevalence of moderate-to-severe depression (rPR = 3.21), anxiety (rPR = 4.13), and stress (rPR = 1.19) (Table 3). Similarly, having two or more symptoms of COVID-19 over the previous two weeks increased the prevalence of moderate-to-severe anxiety (rPR = 2.59), stress (rPR = 1.16), and depression (rPR = 4.36). These last values were maintained after adjusting for other variables (aPR = 3.59). Meanwhile, living alone was a protective factor for the presence of moderate-to-severe depression (rPR = 0.49) and anxiety (rPR = 0.65).

The adjusted models indicated that having a history of mental health problems increases the chance of moderate-to-severe depression (aPR = 3.34) and anxiety (aPR = 3.42) more than three-fold. Similarly, worrying about losing employment also increases the chance of moderate to severe depression (aPR = 3.39) and anxiety (aPR = 3.62) more than threefold (Table 3). Being afraid of infecting family members also increases the chance of having moderate-to-severe depression (aPR = 2.91) and anxiety (aPR = 2.51). However, according to the proposed manual forward regression, only the covariate type of healthcare worker would have been included in the adjusted regression model for this outcome, so we only present the results from the raw regression model.

### 3.2. Qualitative Results

Among the 20 hospital workers interviewed (Table 4), a qualitative analysis identified five recurring topics related to the impact of the pandemic on the worker’s life at a personal level.

#### 3.2.1. Emotional Distress Linked to Hospital Experiences of Suffering and Death

Health workers, especially clinicians, stated that they experienced emotional distress during the pandemic, directly relating it to their proximity to hospital experiences of suffering and death in the pandemic context.

“It was shocking; it was really difficult to see 15 patients die every day. I don’t know if you have seen the rubbish packages in the streets, in black bags, that’s how the corpses were. Containers full. In truth it was like that in all [hospital] shifts.”(SR.03.09.10)

#### 3.2.2. Fear of Contracting COVID-19

Among the most frequent reports, we found the fear of contracting COVID-19. This was linked to the exposure to the virus in the hospital environment, lack of personal protective equipment, heavy workloads with COVID-19 patients, and contaminated environments. It was also linked to the high rates of morbidity and mortality in health workers, and individual characteristics of the workers and their families in relation to a predisposition as a member of a population at risk (age, previous physical illnesses).

“I don’t really want to go to work. I’m scared, I’m scared, I pray all the time from here [hospital] to my house. I go praying and I say God please take care of me.”(AV.04.10.10)

“In those months that all this began [COVID-19 pandemic], in April, May, until June I felt terrible fear; and I think most of my colleagues too. It was a terror to come and work and when they told me “Yes, the patient you treated came out positive” we were terrified, we were in panic, I really did not know what to do.”(AV.05.10.10)

#### 3.2.3. Modification of Life Routines

Health workers modified their daily life routines, restructuring their working hours, requesting unpaid leave, avoiding public transportation, and even avoiding contact with their families.

“Since I have restarted the work in July [2020], it is no longer the same. For example, I have to take a private mobility to go to work. Until now, I have not gotten on a bus because of the fear of getting infected.”(AV.01.07.10)

#### 3.2.4. Exacerbation of Mental Disorders

A group of workers with pre-existing mental health diagnoses (e.g., depression and anxiety) reported that the context of the pandemic and the conditions of hospital work exacerbated their symptoms.

“The change I feel is very marked… I feel affected with a depression that has become more and more profound. And the little tolerance for hospital work. So, I imagine that, like me, there are many colleagues, or many people close to the health system who have had this type of condition as well.”(SR.04.11.10)

“On a personal level, there is more anxiety, more fear, now for example I am medicated, I am on alprazolam at night because it was not my circadian rhythm to sleep every day at the same time.”(SR.03.11.10)

#### 3.2.5. Physical Problems Associated with Emotional Distress

Some workers related the appearance of physical symptoms to their state of emotional distress, such as the presence of insomnia, tachycardia, changes in their eating habits (lack of or increase in appetite), weight loss, gastric discomfort, and dyspnea.

“At least in my case, I have anxiety attacks. At first, it started giving me like… episodes of sinus tachycardia and I went to the cardiologist. That had never happened to me, never in my life that had happened to me. The cardiologist did an ultrasound on me, did pertinent tests and everything and said: “You’re fine. I think you have to start regulating the work issue a bit, lower the intensity with what you do because that is affecting you.” And up to now I am still on medications.”(AV.03.10.10).

## 4. Discussion

### 4.1. Main Findings and Significance of the Results

In this mixed-methods cross-sectional study, we found changes in the mental health of the hospital workers in both the quantitative and qualitative data. The participants reported a high prevalence of anxiety symptoms (28.2%), depressive symptoms (29.2%), and perceived stress (78.5%). These results were similar to those reported in a recent meta-analysis of studies of healthcare workers, mainly performed in Asia and South America, which found a high prevalence of anxiety symptoms (26.0%), depressive symptoms (25.0%), and stress (40.0%) [27]. They were also similar to the results of a recent study conducted in Peruvian clinical healthcare workers [41]. Healthcare workers are experiencing similar effects on their mental health during the COVID-19 pandemic all over the world.

We identified some sociodemographic variables related to a higher prevalence of mental health problems. Women had higher frequencies of moderate-to-severe stress and depression, as reported previously [12,42]. This widespread result may be rooted in a variety of causes, including sex hormone differences, gender-based violence, and social and healthcare discrimination [43]. Studies conducted before the pandemic indicate that living alone is a risk factor for negative effects on mental health in the general population [44,45] and in healthcare workers [46,47]. However, we found that healthcare workers living alone had a lower frequency of mental health problems, whereas living with a person at risk of poorer outcomes from COVID-19 infection was linked to a greater incidence of poor outcomes in mental health. This finding could be a result of the fear that hospital workers could have about infecting their relatives. This may cause poor mental health; recognizing this, some mental health workers may have decided to live alone in order to reduce or eliminate that fear.

Healthcare workers who had a previous diagnosis of mental health problems had worse outcomes for depressive symptoms, anxiety symptoms, and perceived stress. This result was plausible because the COVID-19 pandemic can cause reactive mental health symptoms, which along with working inside hospitals, can exacerbate the mental health issues of the healthcare workers with a previous mental health diagnosis or a history of psychiatric treatment [48]. This subpopulation is especially vulnerable to stressors and should promptly receive mental healthcare attention with adequate follow-up.

During the COVID-19 pandemic, we must consider the stigma against hospital workers. It is noticeable that a perception of frequent stigma for being hospital staff is related to a greater prevalence of depressive symptoms, anxiety symptoms, and perceived stress. The general population may consider that hospital workers (clinical or non-clinical) may be vehicles of COVID-19 and exclude them from the rest of the society. This exclusion could trigger feelings of depression and futility, which took a significant psychological toll in healthcare workers [49]. During other epidemics and/or pandemics, studies indicated that 20–49% of the healthcare workers experienced social stigma. A survey of 187 nurses during the MERS-CoV outbreak in South Korea found that the feelings of stigma directly or indirectly influenced their mental health [50]. Zandifar et al. found a significant and robust correlation during COVID-19 between the perception of stigma and post-traumatic stress disorder symptoms in healthcare workers in Iran [49]. Regardless of the role, whether as a clinical or nonclinical worker, those who work at a hospital may experience stigma, especially women, first-line workers, physicians, and residents [49]. This feeling should be noted during psychological or psychiatric treatment.

Self-reported fear of losing employment was positively associated with a greater prevalence of depressive and anxiety symptoms. These results are in line with those obtained by Wilson et al. who found that greater job insecurity due to COVID-19 among those currently employed is related to greater depressive symptoms [51]. Financial concerns help to explain why those with greater job insecurity report higher levels of anxiety symptoms [51]. The pandemic took a toll on the healthcare system and the economy of our country. Peru has among the highest rate of COVID-19 incidence, and many Peruvians lost their employment in 2020 [52]. Peruvian healthcare personnel from public hospitals have had difficulty securing permanent positions [53], and their labor situation is unstable [54]. The ever-present possibility of losing employment may result in an increase in mental symptoms [55]. The authorities should be conscious of this and ensure—as much as possible—labor stability in all hospital workers, regardless of their clinical or nonclinical status.

In the qualitative data, we found, echoing the results of other investigations, that healthcare workers were exposed to a range of conditions associated with the COVID-19 pandemic that are harmful to mental health [56]. Kotera et al. in their qualitative study in Japanese healthcare workers reported that levels of stress and loneliness were increased, while their coping strategies were limited. Intrinsic rewards such as workplace communication and acknowledgment of their work were identified as positive resources for their mental health [57]. In a qualitative systematic review involving thematic synthesis revealed the burden of healthcare workers providers during the COVID-19 pandemic, the four main themes are: inadequate preparedness; emotional challenges; insufficient equipment and information; and work burnout [58]. In our study, the participants reported forms of stress such as trauma associated with the COVID-19 related deaths, modifications to their life routine, and fear of the COVID-19. We found topics related to the changes in workers’ lives at a personal level, including emotional distress linked to hospital experiences of suffering and death. Work-related stress is a cause for concern in healthcare workers. It has been associated with anxiety, including multiple clinical effects, such as depression due to seeing countless numbers of deaths and work overload [59]. Physical problems associated with emotional distress from the pandemic included frequent somatic symptoms caused by significant work-related psychological pressure in professionals who were directly involved in the care of patients infected with COVID-19 [60]. Fear is among the most influential factors on mental health problems, including anxiety and stress. The triad of fear, anxiety, and post-traumatic stress disorder may explain over 70% of depressive symptoms in the general population and among healthcare workers during the COVID-19 pandemic [61].

### 4.2. Clinical Relevance

Our study identified two findings for the study of the mental health of healthcare workers. First, we found that job instability (fear of losing one’s job) had a major impact on the mental health of healthcare workers. This is very relevant, as different countries have created a large number of temporary positions within their healthcare facilities to cope with the COVID-19 pandemic, which, although intended as a rapid response to the fight against the lack of personnel caused by the virus infections, represents a risk factor for the mental health of healthcare workers. Secondly, face-to-face work at the beginning of the pandemic represented a cause of stress and fear on most of the healthcare personnel, this happened because the nature of the work did not allow them to use virtual means. Therefore, opting for the non-face-to-face options (when possible) could be a good strategy to improve workers’ mental health.

This study indicated a high prevalence of mental symptoms in hospital workers. It is urgent to develop psychological intervention plans based on interdisciplinary teams and consider implementing telemedicine options for mental care attention. Early psychological interventions targeting this vulnerable group may be beneficial [62]. A study at a hospital in Hunan province in China suggests several means of preventing mental health problems in healthcare workers. These included, but were not limited to, installing rest spaces in which workers can temporarily isolate themselves from their families, having adequate food and access medical supplies available daily, and providing adequate information about the course of the disease and the pandemic regularly, along with good protection measures, developing detailed and clear rules for the use and management of personal protection equipment, offering counseling in techniques of relaxation and stress management, and bringing psychologists to the rest areas to listen to the healthcare workers’ difficulties and providing necessary support [63]. In addition, it is suggested that health policies aimed at implementing various mental health services be implemented. These policies include screenings with standardized online evaluations, educational interventions in mental health, provision of psychological support after the detection of vulnerable patients, and adequate psychiatric care for mental health management. All of these measures will empower Peru in the containment and future eradication of the COVID-19 pandemic [64].

Furthermore, workers’ confidence is a crucial aspect for their psychological wellbeing because its presence improves their motivation, performance, and attention capacity. The experience of the H1N1 pandemic in Japan in 2009 suggests that workers’ confidence in their peers is an important element for being willing to work during a public health crisis. This promotes better social interactions and cooperation among healthcare workers [65]. In Peru, the Health Ministry published a guideline for the mental healthcare of healthcare workers during the COVID-19 pandemic. It shows a flow diagram suggesting that after the symptoms of mental health problems in healthcare workers are identified, they must receive psychosocial treatment [66]. However, these instructions do not specify how these consultations should be performed or whether this can be carried out remotely. In addition, this guideline excludes nonclinical healthcare workers such as security and cleaning staff. Our study recommends including nonclinical healthcare workers inside the mental healthcare policies.

### 4.3. Limitations and Future Studies

This study must be understood in relation to its methodological limitations. First, convenience sampling does not allow an adequate statistical potency for some associations. Nevertheless, these results are important because they provide data on different factors associated with mental health problems, and the mixed design provides additional understanding, explanations, and interpretations of the quantitative results. However, beyond nonstatistical significance, we cannot rule out certain associations. We considered it necessary not to use indicators such as mean or standard deviation, since we categorized our outcomes (ordinal or dichotomous) in order to facilitate their interpretation. It should be noted that this is a common practice in observational studies. Second, we cannot assess any causal relationships nor establish how different mental health outcomes will evolve. Future research should examine how the levels of anxiety, depression, and stress are changing in broader Peruvian populations. Finally, we only included workers from two hospitals, so we cannot generalize our results. However, previous reports have found roughly similar results in different countries [27,41,49,67]. Additionally, our study included clinical and nonclinical healthcare workers (security and cleaning staff), the latter of whom are often omitted from studies of this kind even though they are essential workers involved in healthcare. Future studies could evaluate how other occupational factors may have an impact on mental health during pandemic contexts in healthcare workers, such as access to personal protective equipment, workplace harassment, or self-efficacy to deliver bad news to patients in non-healthcare personnel, such as security personnel.

## 5. Conclusions

This study explored the experiences of workers from two tertiary hospitals in Peru. Our results demonstrate that during the COVID-19 pandemic, a high prevalence of depressive symptoms, anxiety symptoms, and perceived stress was observed in clinical and nonclinical healthcare workers. The experience of stigma due to working at a hospital, having a history of mental health problems, worrying about losing employment, being afraid of infecting a family member, and having two or more symptoms of COVID-19 in previous weeks increased the prevalence of having clinically relevant depressive symptoms, anxiety symptoms, and perceived stress. It is necessary to promote mental wellbeing and adequate measures to manage emotional distress, combined with better support for healthcare workers.

Our study makes three recommendations. First, the working conditions and occupational safety of clinical and nonclinical healthcare workers must be improved. Second, assessment and care must be delivered to clinical and nonclinical healthcare workers to improve their mental health. These interventions may need to be carried out using telemedicine to avoid the risk of contagion and exposure to the virus. Third, clinical and nonclinical healthcare workers must be considered beneficiaries of mental health policies and guidelines.

## Figures and Tables

**Figure 1 ijerph-19-05346-f001:**
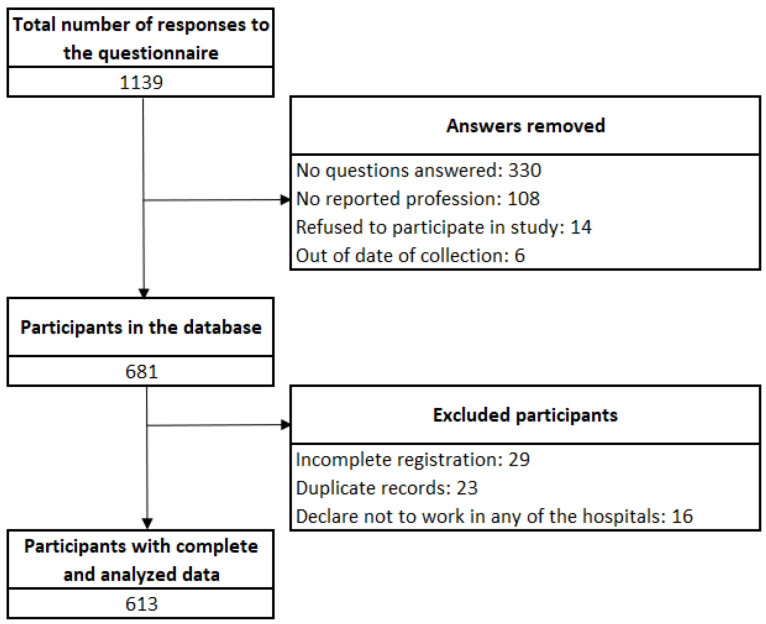
Selection flowchart.

**Table 1 ijerph-19-05346-t001:** Characteristics of participants (*n* = 613).

Characteristics		*n* (%)
Sex	Female	379 (61.8)
	Male	234 (38.2)
Age (years) *		39 (30–47)
Civil status	Single	308 (50.2)
	Married or living together	265 (43.2)
	Divorced	28 (4.6)
	Widowed	12 (2.0)
Hospital	Guillermo Almenara Irigoyen NH.	384 (62.6)
	Edgardo Rebagliati Martins NH.	229 (37.4)
Type of healthcare worker	Physicians	87 (14.2)
	Nurses	122 (19.9)
	Other clinical workers	96 (15.7)
	Non-clinical workers	308 (50.2)
Professed religion		412 (67.2)
Living alone		85 (13.9)
Living with a person at risk		267 (43.6)
Mental health history		42 (6.9)
Anxiety	Normal	440 (71.8)
	Mild	120 (19.6)
	Moderated	40 (6.5)
	Severe	13 (2.1)
Depression	Normal	434 (70.8)
	Mild	124 (20.2)
	Moderated	30 (4.9)
	Moderately severe or Severe	25 (4.1)
Stress	Mild	132 (21.5)
	Moderated	459 (74.9)
	Severe	22 (3.6)

Note: * Medium and interquartile range. NH = National Hospital.

**Table 2 ijerph-19-05346-t002:** Bivariate association between variables and anxiety, stress, and depression levels (*n* = 613).

Characteristics	Anxiety, *n* (%)		*p* Value *	Stress, *n* (%)		*p* Value *	Depression, *n* (%)		*p* Value *
	Normal-to-Mild	Moderate-to-Severe		Mild	Moderate-to-Severe		Normal-to-Mild	Moderate-to-Severe	
**Sex**			0.211			0.002			0.772
Female	342 (90.2)	37 (9.8)		66 (17.4)	313 (82.6)		344 (90.8)	35 (9.2)	
Male	218 (93.2)	16 (6.8)		66 (28.2)	168 (71.8)		214 (91.5)	20 (8.6)	
**Age**			0.437			0.009			0.432
18 to 29 years old	120 (90.9)	12 (9.1)		41 (31.1)	91 (68.9)		116 (87.9)	16 (12.1)	
30 to 39 years old	181 (91.4)	17 (8.6)		43 (21.7)	155 (78.3)		183 (92.4)	15 (7.6)	
40 to 49 years old	153 (93.9)	10 (6.1)		31 (19.0)	132 (81.0)		151 (92.6)	12 (7.4)	
50 years or more	106 (88.3)	14 (11.7)		17 (14.2)	103 (85.8)		108 (90.0)	12 (10.0)	
**Civil status**			0.752			0.543			0.825
Single, divorced, or widowed	319 (91.7)	29 (8.3)		78 (22.4)	270 (77.6)		316 (90.8)	32 (9.2)	
Married or cohabiting	241 (90.9)	24 (9.1)		54 (20.4)	211 (79.6)		242 (91.3)	23 (8.7)	
**Type of healthcare worker**			0.297			<0.001			0.061
Nonclinical workers	285 (92.5)	23 (7.5)		91 (29.6)	217 (70.5)		287 (93.2)	21 (6.8)	
Clinical workers	275 (90.2)	30 (9.8)		41 (13.4)	264 (86.6)		271 (88.9)	34 (11.2)	
**Professed religion**			0.180			<0.001			0.130
No	188 (93.5)	13 (6.5)		60 (29.9)	141 (70.2)		188 (93.5)	13 (6.5)	
Yes	372 (90.3)	40 (9.7)		72 (17.5)	340 (82.5)		370 (89.8)	42 (10.2)	
**Live alone**			0.329			0.630			0.138
No	480 (90.9)	48 (9.1)		112 (21.2)	416 (78.8)		477 (90.3)	51 (9.7)	
Yes	80 (94.1)	5 (5.9)		20 (23.5)	65 (76.5)		81 (95.3)	4 (4.7)	
**Live with a person at risk**			0.022			0.023			0.002
No	324 (93.6)	22 (6.4)		86 (24.9)	260 (75.1)		326 (94.2)	20 (5.8)	
Yes	236 (88.4)	31 (11.6)		46 (17.2)	221 (82.8)		232 (86.9)	35 (13.1)	
**Mental health history**			<0.001			0.006			<0.001
No	534 (93.5)	37 (6.5)		130 (22.8)	441 (77.2)		532 (93.2)	39 (6.8)	
Yes	26 (61.9)	16 (38.1)		2 (4.8)	40 (95.2)		26 (61.9)	16 (38.1)	
**Feeling stigma for working in the hospital**			<0.001			0.018			<0.001
Less than half the days	525 (93.1)	39 (6.9)		128 (22.7)	436 (77.3)		521 (92.4)	43 (7.6)	
More than half of the days	35 (71.4)	14 (28.6)		4 (8.2)	45 (91.8)		37 (75.5)	12 (24.5)	
**Fear of infecting family members**			<0.001			<0.001			<0.001
Less than half the days	337 (96.6)	12 (3.4)		94 (26.9)	255 (73.1)		336 (96.3)	13 (3.7)	
More than half of the days	223 (84.5)	41 (15.5)		38 (14.4)	226 (85.6)		222 (84.1)	42 (15.9)	
**Thinking about losing your job**			<0.001			0.001			<0.001
Less than half the days	514 (94.7)	29 (5.3)		128 (23.6)	415 (76.4)		511 (94.1)	32 (5.9)	
More than half of the days	46 (65.7)	24 (34.3)		4 (5.7)	66 (94.3)		47 (67.1)	23 (32.9)	
**Having two or more symptoms of COVID-19 in the last few weeks**			<0.001			0.002			<0.001
No	428 (93.9)	28 (6.1)		112 (24.6)	344 (75.4)		434 (95.2)	22 (4.8)	
Yes	132 (84.1)	25 (15.9)		20 (12.7)	137 (87.3)		124 (79.0)	33 (21.0)	

Note: * Chi-square test. Mod-Severe = moderate-to-severe.

**Table 3 ijerph-19-05346-t003:** Forward regression raw and adjusted models for associated variables with moderate-to-severe anxiety, stress, and depression (*n* = 613).

Characteristics	Moderate-to-Severe Anxiety		Moderate-to-Severe Stress	Moderate-to-Severe Depression	
	rPR (95% CI)	aRP (95%CI) *	rPR (95% CI) **	rPR (95% CI)	aRP (95%CI) ***
**Sex**					
Male	Ref.	–	Ref.	Ref.	–
Female	1.43 (0.62–3.29)		1.15 (1.07–1.24)	1.08 (1.00–1.17)	
**Age**					
18 to 29 years old	Ref.	–	Ref.	Ref.	Ref.
30 to 39 years old	0.94 (0.65–1.36)		1.14 (0.94–1.37)	0.63 (0.42–0.93)	0.34 (0.15–0.77)
40 to 49 years old	0.67 (0.23–2.01)		1.17 (1.08–1.27)	0.61 (0.37–0.99)	0.63 (0.47–0.85)
50 years or more	1.28 (1.18–1.40)		1.25 (1.20–1.29)	0.83 (0.31–2.19)	0.70 (0.17–2.88)
**Civil status**					
Single, divorced, or widowed	Ref.	–	Ref.	Ref.	–
Married or cohabiting	1.09 (1.08–1.10)		1.03 (1.03–1.03)	0.94 (0.69–1.28)	
**Type of healthcare worker**					
Nonclinical workers	Ref.	–	Ref.	Ref.	–
Clinical workers	1.32 (0.76–2.28)		1.23 (1.13–1.34)	1.63 (1.11–2.42)	
**Professed religion**					
No	Ref.	–	Ref.	Ref.	–
Yes	1.50 (1.39–1.62)		1.18 (1.04–1.33)	1.58 (0.73–3.42)	
**Living alone**					
No	Ref.	–	Ref.	Ref.	–
Yes	0.65 (0.59–0.70)		0.97 (0.85–1.10)	0.49 (0.38–0.63)	
**Living with a person at risk of COVID-19**					
No	Ref.	–	Ref.	Ref.	–
Yes	1.83 (1.09–3.06)		1.10 (1.03–1.17)	2.27 (1.53–3.36)	
**Mental health history**					
No	Ref.	Ref.	Ref.	Ref.	Ref.
Yes	5.88 (4.84–7.14)	3.42 (2.97–3.94)	1.23 (1.23–1.24)	5.58 (4.71–6.61)	3.34 (2.90–3.84)
**Feeling stigma for working in the hospital**					
Less than half the days	Ref.	–	Ref.	Ref.	–
More than half of the days	4.13 (1.80–9.50)		1.19 (1.12–1.25)	3.21 (2.75–3.75)	
**Fear of infecting family members**					
Less than half the days	Ref.	Ref.	Ref.	Ref.	Ref.
More than half of the days	4.52 (4.18–4.89)	2.91 (2.75–3.08)	1.17 (1.11–1.23)	4.27 (2.90–6.28)	2.51 (1.31–4.80)
**Thinking about losing their job**					
Less than half the days	Ref.	Ref.	Ref.	Ref.	Ref.
More than half of the days	6.42 (4.23–9.75)	3.62 (2.04–6.43)	1.23 (1.15–1.32)	5.58 (4.92–6.32)	3.39 (2.62–4.27)
**Having two or more symptoms of COVID-19 in the last few weeks**					
No	Ref.	–	Ref.	Ref.	Ref.
Yes	2.59 (1.62–4.15)		1.16 (1.16–1.16)	4.36 (3.70–5.13)	3.59 (2.11–6.09)

Note: rPR = raw prevalence ratio. aPR = adjusted prevalence ratio. 95% CI = 95% confidence interval. * Model adjusted for identified covariates using forward regression analysis: mental health history, fear of infecting family members, and thinking of losing their job. ** Only data for raw regression model is presented, because according to the proposed manual forward regression, only the covariate type of healthcare worker would have entered in the adjusted regression model for this outcome. *** Model adjusted for identified covariates using forward regression analysis: age, mental health history, fear of infecting family members, thinking of losing their job, and having two or more symptoms.

**Table 4 ijerph-19-05346-t004:** Characteristics of the hospital workers interviewed for the qualitative analysis (*n* = 20).

Age	Sex	Work Area	Symptoms
**Clinical Workers**			
32	Female	Hospitalization	Moderate anxiety
29	Female	Hospitalization	Severe depression
53	Male	Emergency	Moderate stress
28	Female	ICU	Severe stress
47	Female	Hospitalization	Moderate anxiety
34	Male	ICU	Severe depression
60	Female	Hospitalization	Severe stress
37	Female	Hospitalization	Severe stress
**Nonclinical workers**			
39	Male	Administrative	Moderate anxiety
40	Female	Administrative	Moderate-to-severe depression
58	Female	Administrative	Severe stress
30	Male	Administrative	Moderate-to-severe depression
47	Female	Cleaning	Moderate anxiety
51	Female	Cleaning	Moderate-to-severe depression
38	Female	Cleaning	Severe stress
23	Male	Cleaning	Severe stress
26	Male	Security	Severe anxiety
54	Female	Security	Moderate-to-severe depression
40	Male	Security	Moderate stress
40	Male	Security	Moderate stress

## Data Availability

The data presented in this study are available on request from the corresponding author.

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
