# Peer review of "Factors Associated with Mental Health Outcomes in Hospital Workers during the COVID-19 Pandemic: A Mixed-Methods Study"

_ijerph, 2022, doi:10.3390/ijerph19095346_

Round 1
Reviewer 1 Report
The article has been improved, but still warrants some improvements.
Since the first review more articles have come out on the topic. Update the literature. Some suggested:
Khudaykulov, A., Changjun, Z., Obrenovic, B. et al. The fear of COVID-19 and job insecurity impact on depression and anxiety: An empirical study in China in the COVID-19 pandemic aftermath. Curr Psychol (2022). https://doi.org/10.1007/s12144-022-02883-9
Hamaideh, S. H., Al‐Modallal, H., Tanash, M. A., & Hamdan‐Mansour3, A. (2022). Depression, anxiety and stress among undergraduate students during COVID‐19 outbreak and" home‐quarantine". Nursing Open, 9(2), 1423-1431.It is mandatory to address the comments of the reviewer regarding the results in the limitations and future studies section.
"We considered it necessary not to use indicators such as mean or standard deviation, since we categorized our outcomes (ordinal or dichotomous), in order to facilitate their interpretation. It should be noted that this is a common practice in observational studies."
3. There are many important errors in the data analysis. A continuous variable should be analyzed as continuous, since information is lost and the conclusions are not correct - for example, stress is divided between mild (0-13) and moderate to severe (14-41). Two people who have 13 and 14 points respectively on the scale are considered in the same category as a person who has 0 and 41 respectively, it does not make sense. The same applies to the rest of the scales. The whole results section must be rewritten according to these requests.
Repy"In numerous studies, the scales we used were categorized as clinically relevant and clinically not relevant. We have added the bibliographic references that support this point. Because of this, we do not consider it necessary to rewrite the entire results section."
Author Response
Reviewer 1
The article has been improved, but still warrants some improvements.
Since the first review more articles have come out on the topic. Update the literature. Some suggested:
Khudaykulov, A., Changjun, Z., Obrenovic, B. et al. The fear of COVID-19 and job insecurity impact on depression and anxiety: An empirical study in China in the COVID-19 pandemic aftermath. Curr Psychol (2022). https://doi.org/10.1007/s12144-022-02883-9
Hamaideh, S. H., Al‐Modallal, H., Tanash, M. A., & Hamdan‐Mansour3, A. (2022). Depression, anxiety and stress among undergraduate students during COVID‐19 outbreak and" home‐quarantine". Nursing Open, 9(2), 1423-1431.
Reply: References suggested by the reviewer were added.
It is mandatory to address the comments of the reviewer regarding the results in the limitations and future studies section.
"We considered it necessary not to use indicators such as mean or standard deviation, since we categorized our outcomes (ordinal or dichotomous), in order to facilitate their interpretation. It should be noted that this is a common practice in observational studies."
Reply: Comments suggested by the reviewer were added in the recommended section.
There are many important errors in the data analysis. A continuous variable should be analyzed as continuous, since information is lost and the conclusions are not correct - for example, stress is divided between mild (0-13) and moderate to severe (14-41). Two people who have 13 and 14 points respectively on the scale are considered in the same category as a person who has 0 and 41 respectively, it does not make sense. The same applies to the rest of the scales. The whole results section must be rewritten according to these requests.
Repy: In numerous studies, the scales we used were categorized as clinically relevant and clinically not relevant. We have added the bibliographic references that support this point. Because of this, we do not consider it necessary to rewrite the entire results section.
Reviewer 2 Report
The authors of this manuscript took up a very important topic in the fieldof Factors relating to the mental health outcomes of workers in two Peruvian
hospitals during the COVID-19 pandemic: A mixed-method study. The reviewer is very impressed with how perfectly the authors have transformed
/ modified the previous version of the manuscript into the present one.
The findings of the authors are valuable in terms of the perception of frequent
stigmatization of hospital staff, more frequent occurrence of depressive symptoms,
anxiety symptoms and perceived stress. According to the authors, the general population
may believe that hospital staff (clinical or non-clinical) may be COVID-19 vehicles and
exclude them from the rest of society. A wonderful method was used by the authors of this work to cite the statements of
participants in the study, confirming not only their state of mental exhaustion,
but also showing how much work in the health care system destroys many aspects
of their private lives during the COVID-19
pandemic. In the opinion of the reviewer and for a better understanding of the text
by the reader, I recommend that the list of references be modified
in accordance with the journal's guidelines. The scientific sound of the manuscript is valuable and this text deserving
of publication.
Author Response
Reviewer 2
The authors of this manuscript took up a very important topic in the field
of Factors relating to the mental health outcomes of workers in two Peruvian
hospitals during the COVID-19 pandemic: A mixed-method study. The reviewer is very impressed with how perfectly the authors have transformed/ modified the previous version of the manuscript into the present one. The findings of the authors are valuable in terms of the perception of frequent stigmatization of hospital staff, more frequent occurrence of depressive symptoms, anxiety symptoms and perceived stress. According to the authors, the general population may believe that hospital staff (clinical or non-clinical) may be COVID-19 vehicles and exclude them from the rest of society. A wonderful method was used by the authors of this work to cite the statements of participants in the study, confirming not only their state of mental exhaustion,
but also showing how much work in the health care system destroys many aspects of their private lives during the COVID-19 pandemic.
In the opinion of the reviewer and for a better understanding of the text
by the reader, I recommend that the list of references be modified
in accordance with the journal's guidelines. The scientific sound of the manuscript is valuable and this text deserving
of publication.
Reply: Thanks to the reviewer for their comments. The references have been adjusted according to the format of the journal.
Reviewer 3 Report
Dear colleagues, I hope this message find you well.
Thank you for giving me the opportunity of reading the work “Factors associated with mental health outcomes in workers from two Peruvian hospitals during the COVID-19 pandemic: A mixed-methods study”, it has been a very big pleasure to collaborate reviewing this manuscript. The topic of this paper is very interesting and it seems necessary to delve it. However, there are several questions to improve before to publish it. I would suggest some changes:
Title and abstract
- Ok
Introduction
- Dear colleagues, the structure of the introduction is not clear. I recommended to divide the introduction into several subsections. For example, creating a specific subsection where describe each group of variables/factors involved.
- Secondly, when you explain the signs of psychological distress and mental health as a result of COVID-19 (page 2), it is necessary to add more data and references. I recommend you to add this paper recently published (10.1016/S2215-0366(20)30089-4), which proposes COVID-19 pandemic as a PTSD.
- Moreover, at lines 86-104, introducing more studies could be interesting in order to support better how COVID-19 has psychologically affected health-care professionals:
https://doi.org/10.1016/j.jaac.2020.08.466
https://doi.org/10.1016/S2215-0366(20)30307-2
- Aims and hypotheses should be inside of introduction section.
Method
- Sample items should be added in each questionnaire.
Results
- After the first review process, results have been improved significantly, congratulations.
Discussion
- In my humble opinion, it could be useful to describe in more detail the practical and theoretical implications of this research. It would be useful they contextualize better the contribution within the framework of the issue explaining why the contribution is useful and enrich the impact.
Conclusions
- Nothing to add. Good job.
Author Response
Reviewer 3
Dear colleagues, I hope this message find you well.
Thank you for giving me the opportunity of reading the work “Factors associated with mental health outcomes in workers from two Peruvian hospitals during the COVID-19 pandemic: A mixed-methods study”, it has been a very big pleasure to collaborate reviewing this manuscript. The topic of this paper is very interesting and it seems necessary to delve it. However, there are several questions to improve before to publish it. I would suggest some changes:
Title and abstract
- Ok
Introduction
- Dear colleagues, the structure of the introduction is not clear. I recommended to divide the introduction into several subsections. For example, creating a specific subsection where describe each group of variables/factors involved.
Reply: Thanks for your recomendation. We create subsections: 1.1.Mental health in heatlhcare workers of COVID-19 patients, 1.2.Factors associated with mental health in heatlhcare workers, 1.3.Aims of the investigation
- Secondly, when you explain the signs of psychological distress and mental health as a result of COVID-19 (page 2), it is necessary to add more data and references. I recommend you to add this paper recently published (10.1016/S2215-0366(20)30089-4), which proposes COVID-19 pandemic as a PTSD.
Reply: Thanks for your recomendation. We have added the following paragraph in the introduction: “COVID-19 pandemic is a new type of trauma that have never been conceptually or empirically analyzed in the field of mental health and psychiatry research [6]. Different factors make COVID-19 a unique trauma type: a) it is a constant and permanent traumatic stress, b) it is a multiple complex trauma, and c) is not necessarily related to the actual infection of COVID-19, but also is more related to the perceived threat of the uncontrolled virus and the direct and indirect social consequences [6]. COVID-19 pandemic can be understood as a traumatic stressor capable of exacerbating mental health problems [7].”
- Moreover, at lines 86-104, introducing more studies could be interesting in order to support better how COVID-19 has psychologically affected health-care professionals:
https://doi.org/10.1016/j.jaac.2020.08.466
https://doi.org/10.1016/S2215-0366(20)30307-2
Reply: References suggested by the reviewer were added.
- Aims and hypotheses should be inside of introduction section.
Reply: In the final part of the introduction we place the objective of the study: “we developed a sequential quantitative-qualitative mixed-methods study design to examine the following research questions. 1. What is the prevalence of mental health problems in a sample of workers from two tertiary hospitals in Lima, Peru? 2. What factors are associated with positive or negative mental health outcomes in this sample? 3. What are the workers’ perceptions and beliefs about the impact of the pandemic on their mental health?”
Method
- Sample items should be added in each questionnaire.
Reply: Example items were added in each questionnaire.
Results
- After the first review process, results have been improved significantly, congratulations.
Reply: Thanks to the reviewer for his kind words.
Discussion
- In my humble opinion, it could be useful to describe in more detail the practical and theoretical implications of this research. It would be useful they contextualize better the contribution within the framework of the issue explaining why the contribution is useful and enrich the impact.
Reply: Thanks for your recomendation. We have added the following paragraph: “In addition, it is suggested that health policies aimed at implementing various mental health services be implemented. These policies include screenings with standardized online evaluations, educational interventions in mental health, provision of psychological support after the detection of vulnerable patients, and adequate psychiatric care for mental health management. All of these measures will empower Peru in the containment and future eradication of the COVID-19 pandemic [64].”
Conclusions
- Nothing to add. Good job.
Reply: Thanks to the reviewer for his kind words.
This manuscript is a resubmission of an earlier submission. The following is a list of the peer review reports and author responses from that submission.
Round 1
Reviewer 1 Report
The reviewer read with enthusiasm this valuable work focused on, inter alia, on non-clinical healthcare workers who are expose to the same factors as clinical workers, but non-clinical healthcare workers are not as often included in other scientific publications on the COVID-19 pandemic outcomes.
Taking into consideration the instrument used in the research, one may doubt : what was the purpose of including questions about the participant's religion and marital status in such a survey? What was the relevance of these confidential questions to factors related to mental health outcomes during the COVID-19 pandemic?
An excellent method had been used by the authors of this work to cite statements of participants taking part in the study, confirming not only their state of mental exhaustion, but also showing how much work in the health care system devastating many aspects of their private lives during the COVID-19 pandemic
The text is valuable and could be recommend for publication. However, explanation and annotation into the text shall be insert about reasons and background necessity to use confidential questions about the participant's religion and marital status in such a survey
Author Response
Reviewer 1
The reviewer read with enthusiasm this valuable work focused on, inter alia, on non-clinical healthcare workers who are expose to the same factors as clinical workers, but non-clinical healthcare workers are not as often included in other scientific publications on the COVID-19 pandemic outcomes.
Reply: Thanks to the reviewer for his kind words.
Taking into consideration the instrument used in the research, one may doubt: what was the purpose of including questions about the participant's religion and marital status in such a survey? What was the relevance of these confidential questions to factors related to mental health outcomes during the COVID-19 pandemic?
Reply: Background on the importance of religion and marital status with the mental health of health workers was added in the introduction: “The identified risk and protective factors were summarized as intrapersonal (sex, age, sleep quality, etc.); interpersonal (marital status, familial support, family history of mental disease, peer support, etc.); and organizational (years of work experience, level of training, job satisfaction, etc.) [1] [1] [1]. In addition to emotional coping strategies, religious faith has an impact on the mental health and level of happiness of healthcare workers during the COVID-19 pandemic.”
An excellent method had been used by the authors of this work to cite statements of participants taking part in the study, confirming not only their state of mental exhaustion, but also showing how much work in the health care system devastating many aspects of their private lives during the COVID-19 pandemic.
Reply: Thanks to the reviewer for his kind words.
The text is valuable and could be recommend for publication. However, explanation and annotation into the text shall be insert about reasons and background necessity to use confidential questions about the participant's religion and marital status in such a survey
Reply: Thanks to the reviewer for his suggestions. Background on the importance of religion and marital status with the mental health of health workers was added in the introduction.
Reviewer 2 Report
Authors analyze the consequences of covid in a sample of health employees in Peru. The subject is important, but there are several issues that need to be addressed for the paper to make a significant contribution to research.
In the first place, a quick search allows us to identify dozens of similar papers published in the last two years. What is new about this paper? What does it contribute the analysis of Peru’s employees? The authors should try to enhance the novelty of this paper; publishing an article on a topic that is already known does not make sense.
In the same way, it is necessary to describe and elaborate in the introduction all the variables that later are analyzed in the paper. A research does not consist of applying regressions of all the variables collected to see what effects have on psychological aspects. variables should be identified and defined based on a theoretical analysis of the problem, which has not been done in the introduction. The introduction should end with predictions of the relationships between the important variables. The results section should begin with a descriptive analysis of the variables (means and standard deviations) and later analyze these predictions
There are many important errors in the data analysis. A continuous variable should be analyzed as continuous, since information is lost and the conclusions are not correct - for example, stress is divided between mild (0-13) and moderate to severe (14-41). Two people who have 13 and 14 points respectively on the scale are considered in the same category as a person who has 0 and 41 respectively, it does not make sense. The same applies to the rest of the scales.
The whole results section must be rewritten according to these requests.
Author Response
Reviewer 2
Authors analyze the consequences of covid in a sample of health employees in Peru. The subject is important, but there are several issues that need to be addressed for the paper to make a significant contribution to research.
In the first place, a quick search allows us to identify dozens of similar papers published in the last two years. What is new about this paper? What does it contribute the analysis of Peru’s employees? The authors should try to enhance the novelty of this paper; publishing an article on a topic that is already known does not make sense.
Reply: We added: “The immediate priority is collecting data on the psychological effects of the COVID-19 pandemic across vulnerable groups, such as healthcare workers [2]. However, non-clinical hospital workers are not as often included in scientific publications on the COVID-19 pandemic outcomes. Furthermore, based on the pressing need to respond to the crisis epidemiological studies have been undertaken, however, less attention has been paid to the contextualized experiences and meanings attributed to COVID-19 and strategies to mitigate its spread on healthcare and non-healthcare workers of hospitals [3].”
In the same way, it is necessary to describe and elaborate in the introduction all the variables that later are analyzed in the paper. A research does not consist of applying regressions of all the variables collected to see what effects have on psychological aspects. variables should be identified and defined based on a theoretical analysis of the problem, which has not been done in the introduction. The introduction should end with predictions of the relationships between the important variables. The results section should begin with a descriptive analysis of the variables (means and standard deviations) and later analyze these predictions
Reply: Thanks to the reviewer for his suggestions. We describe in the introduction the variables.
There are many important errors in the data analysis. A continuous variable should be analyzed as continuous, since information is lost and the conclusions are not correct - for example, stress is divided between mild (0-13) and moderate to severe (14-41). Two people who have 13 and 14 points respectively on the scale are considered in the same category as a person who has 0 and 41 respectively, it does not make sense. The same applies to the rest of the scales.
The whole results section must be rewritten according to these requests.
Reply: In numerous studies, the scales we used were categorized as clinically relevant and clinically not relevant. We have added the bibliographic references that support this point.
PSS-10: “Scores for all items are totaled and categorized into three levels of perceived stress: mild (0–13), moderate (14–26), and severe (27–40) [28]. Low-stress level considered as having no stress, while both moderate and high stress levels were merged as having stress [28].”
PHQ-9: “A score of ≥10 has been recommended as the cut-off score for detecting clinically significant major depressive disorders [4].”
GAD-7: “A score of ≥10 was been recommended as the cut-off score for indicating clinically significant anxiety [5].”
Reviewer 3 Report
The research paper is well composed , with proper methods, and satisfactory coherence.
I would like to suggest following improvements:
1. You need to improve theoretical foundation of the paper. How this topic builds on existing literature, and what contributions does it make, what gaps does it fill? Many studies using exact same scales came out in the meantime. Thus, include some recent relevant studies such as :
"Obrenovic, B., Du, J., Godinic, D., Baslom, M. M. M., & Tsoy, D. (2021). The threat of CoViD-19 and job insecurity impact on depression and anxiety: an empirical study in the USA. Frontiers in psychology, 3162."
Khaoula, B. & Jalal, A. (2021). The Impact of Covid-19 on Higher Education: A Review on the Moroccan Case. International Journal of Innovation and Economic Development, 7(5), 17-26.
Guberina, T. & Wang, A.M. (2021). Entrepreneurial Leadership Impact on Job security and Psychological Well-being during the COVID-19 Pandemic: A conceptual review. International Journal of Innovation and Economic Development, 6(6), 7-18.
and many more
2. Emphisize in the introduction section what is the major contribution of the study, and clearly indicate what other studies lack that yours contributes
3. Proofread the paper, english needs some improvment to bring it to the satisfactory level
4. section 4.3 should be renamed to Limitations and future studies in "limitations and future studies" section include some suggestions for future studies
5. contributions can be a part of discussion and concluson. emphisize the innovative points of the study
6. Methodology section seems to include some redundant information, consider revising. Also is the datacollection primary or secondary? You collected data so should be primary?
7. Paper is ok , but requires more work. Please address all points.
Author Response
Reviewer 3The research paper is well composed , with proper methods, and satisfactory coherence.
I would like to suggest following improvements:
- You need to improve theoretical foundation of the paper. How this topic builds on existing literature, and what contributions does it make, what gaps does it fill? Many studies using exact same scales came out in the meantime. Thus, include some recent relevant studies such as:
"Obrenovic, B., Du, J., Godinic, D., Baslom, M. M. M., & Tsoy, D. (2021). The threat of CoViD-19 and job insecurity impact on depression and anxiety: an empirical study in the USA. Frontiers in psychology, 3162."
Khaoula, B. & Jalal, A. (2021). The Impact of Covid-19 on Higher Education: A Review on the Moroccan Case. International Journal of Innovation and Economic Development, 7(5), 17-26.
Guberina, T. & Wang, A.M. (2021). Entrepreneurial Leadership Impact on Job security and Psychological Well-being during the COVID-19 Pandemic: A conceptual review. International Journal of Innovation and Economic Development, 6(6), 7-18.
and many more
Reply: Thanks for the suggestion. Changes suggested by the reviewer have been made. An update of the bibliographic references has been carried out based on the articles suggested by the reviewer.
- Emphasize in the introduction section what is the major contribution of the study, and clearly indicate what other studies lack that yours contributes
Reply: We added: “The immediate priority is collecting data on the psychological effects of the COVID-19 pandemic across vulnerable groups, such as healthcare workers [2]. However, non-clinical hospital workers are not as often included in scientific publications on the COVID-19 pandemic outcomes. Furthermore, based on the pressing need to respond to the crisis epidemiological studies have been undertaken, however, less attention has been paid to the contextualized experiences and meanings attributed to COVID-19 and strategies to mitigate its spread on healthcare and non-healthcare workers of hospitals [3].”
- Proofread the paper, english needs some improvement to bring it to the satisfactory level
Reply: The article was editing by a professional English traductor.
- section 4.3 should be renamed to Limitations and future studies in "limitations and future studies" section include some suggestions for future studies
Reply: The change suggested by the reviewer was made.
“Future studies could evaluate how other occupational factors may have an impact on mental health during pandemic contexts in health care workers such as access to PPE, workplace harassment, or self-efficacy to deliver bad news to patients in non-health care personnel such as security personnel.”
- contributions can be a part of discussion and conclusion. emphasize the innovative points of the study
Reply: We added: “Our study identified two innovations in the study of the mental health of health care workers. First, we found that job instability (fear of losing one's job) has had a major impact on the mental health of health care workers. This is very relevant, as different countries have created a large number of temporary positions within healthcare facilities to cope with the COVID-19 pandemic, which, although intended as a rapid response to the fight against the virus, represent a risk factor for the mental health of healthcare workers. Secondly, face-to-face work at the beginning of the pandemic represented a generator of stress and fear on the part of healthcare personnel, since due to the nature of the work they could not opt for virtual means. Therefore, opting for non-face-to-face options (when possible) could be a strategy to improve workers' mental health.”
- Methodology section seems to include some redundant information, consider revising. Also is the data collection primary or secondary? You collected data so should be primary?
Reply: The change suggested by the reviewer was made. We have revised the methods section. It has been revise the redundant information. Change “Secondary” to “Primary” was made.
- Paper is ok, but requires more work. Please address all points.
Round 2
Reviewer 2 Report
Dear authors.
I have been reading the new version of the paper and unfortunately any of the comments and doubts suggested in the first revision has been addressed. The novelty of the work is not highlighted, the main variables and the predictions that justify are not described. The result section doesn't change. There is nothing new in this version.
Author Response
1. Authors analyze the consequences of covid in a sample of health employees in Peru. The subject is important, but there are several issues that need to be addressed for the paper to make a significant contribution to research. In the first place, a quick search allows us to identify dozens of similar papers published in the last two years. What is new about this paper? What does it contribute the analysis of Peru’s employees? The authors should try to enhance the novelty of this paper; publishing an article on a topic that is already known does not make sense.
Reply 1: We added: “The immediate priority is collecting data on the psychological effects of the COVID-19 pandemic across vulnerable groups, such as healthcare workers [23]. However, non-clinical hospital workers are not as often included in scientific publications on the COVID-19 pandemic outcomes. Furthermore, based on the pressing need to respond to the crisis, many studies focused on epidemiological and quantitative studies, however, less attention has been paid to the contextualized experiences and meanings attributed to COVID-19 and strategies to mitigate its spread on healthcare and non-healthcare workers of hospitals [24]. Quantitative data analysis can provide a more accurate understanding of the experience of people, resulting in a clearer picture by combining quantitative and qualitative methods of the self-reported effects of the COVID-19 pandemic on the mental health of hospital workers'.”
2. In the same way, it is necessary to describe and elaborate in the introduction all the variables that later are analyzed in the paper. A research does not consist of applying regressions of all the variables collected to see what effects have on psychological aspects. variables should be identified and defined based on a theoretical analysis of the problem, which has not been done in the introduction. The introduction should end with predictions of the relationships between the important variables. The results section should begin with a descriptive analysis of the variables (means and standard deviations) and later analyze these predictions.
Reply: Our study used the manual stepwise-forward regression strategy since the exploratory nature of out primary aim, and because theoretically, all available variables might been associated with the outcomes. Due to low number of variables, this strategy is efficient to choosing a relatively small number of explanatory variables out of high possible combinations (https://journalofbigdata.springeropen.com/articles/10.1186/s40537-018-0143-6).
In addition, in order to justify adding the different variables within the analysis, information on the relevant variables in the analysis was added in the introduction:
“Stigmatization is common in disease outbreaks and pandemic situation. Social stigma against healthcare workers who are taking care of COVID-19 patients is expected [12]. Frontline workers experienced three times more stigma than those who did not work on the frontline [13]. The impact of stigmatization cannot be limited to the psychological well-being of healthcare workers; it can also affect their professional competencies to provide quality care to the patients during the pandemic time [12].
Previous studies have found that perceived job insecurity has consequences not only on the individuals’ financial capacity, but also on their mental health [14]. Employment uncertainty causes fears of poverty and leads to marginalization, stigmatization, and social exclusion [15]. The presence of job insecurity is strong predictor for depression [16]. Additionally, non-clinical healthcare workers may have lower preparedness for contact with COVID-19 patients [17] while being exposed to the same factors as clinical workers [18]. A lack of balance among professional duty, altruism, and constant fear can cause conflicts and cognitive dissonance in many hospital workers [19]. The identified risk and protective factors were summarized as intrapersonal (sex, age, sleep quality, etc.); interpersonal (marital status, familial support, family history of mental disease, peer support, etc.); and organizational (years of work experience, level of training, job satisfaction, etc.) [20]. In addition to emotional coping strategies, religious faith has an impact on the mental health and level of happiness of healthcare workers during the COVID-19 pandemic [21]. A meta-analysis reported a prevalence of anxiety symptoms (26%), depressive (25%), and post-traumatic stress disorder (3%) in medical staff [22]. These alterations impact the mental well-being of workers and their work activity, patient attention, and decision-making process [10]. Consequently, if suppression of the pandemic is the target goal, hospital workers’ mental well-being must be acknowledged [10].”
We considered it necessary not to use indicators such as mean or standard deviation, since we categorized our outcomes (ordinal or dichotomous), in order to facilitate their interpretation. It should be noted that this is a common practice in observational studies.
3. There are many important errors in the data analysis. A continuous variable should be analyzed as continuous, since information is lost and the conclusions are not correct - for example, stress is divided between mild (0-13) and moderate to severe (14-41). Two people who have 13 and 14 points respectively on the scale are considered in the same category as a person who has 0 and 41 respectively, it does not make sense. The same applies to the rest of the scales. The whole results section must be rewritten according to these requests.
Reply: In numerous studies, the scales we used were categorized as clinically relevant and clinically not relevant. We have added the bibliographic references that support this point. Because of this, we do not consider it necessary to rewrite the entire results section.
PSS-10: “Scores for all items are totaled and categorized into three levels of perceived stress: mild (0–13), moderate (14–26), and severe (27–40) [28]. Low-stress level considered as having no stress, while both moderate and high stress levels were merged as having stress [28].”
PHQ-9: “A score of ≥10 has been recommended as the cut-off score for detecting clinically significant major depressive disorders [4].”
GAD-7: “A score of ≥10 was been recommended as the cut-off score for indicating clinically significant anxiety [5].”